# Topical Insulin Eye Drops: Stability and Safety of Two Compounded Formulations for Treating Persistent Corneal Epithelial Defects

**DOI:** 10.3390/pharmaceutics16050580

**Published:** 2024-04-24

**Authors:** Marta Vicario-de-la-Torre, Virginia Puebla-García, Lidia Ybañez-García, José Javier López-Cano, Miriam Ana González-Cela-Casamayor, Marco Brugnera, Bárbara Burgos-Blasco, David Díaz-Valle, José Antonio Gegúndez-Fernández, José Manuel Benítez-del-Castillo, Rocío Herrero-Vanrell

**Affiliations:** 1Innovation, Therapy and Pharmaceutical Development in Ophthalmology (InnOftal) Research Group (UCM 920415), Department of Pharmaceutics and Food Technology, Faculty of Pharmacy (UCM), Plaza Ramón y Cajal s/n, 28040 Madrid, Spainrociohv@ucm.es (R.H.-V.); 2National Ocular Pathology Network (OFTARED), Carlos III Institute of Health, San Carlos Clinical Hospital Institute of Health Research (IdISSC), 28040 Madrid, Spaindiazvalle@gmail.com (D.D.-V.);; 3Department of Pharmacy, San Carlos Clinical Hospital, 28040 Madrid, Spain; 4Ocular Surface and Inflammation Unit, Department of Ophthalmology, San Carlos Clinical Hospital Institute of Health Research (IdISSC), 28040 Madrid, Spain

**Keywords:** topical insulin, eye drops, compounding stability, in vitro tolerance, persistent epithelial corneal defects

## Abstract

Compounded insulin eye drops were prepared at 1 IU/mL from commercially available subcutaneous insulin by dilution in saline solution or artificial tears. Physicochemical characterization and in vitro tolerance testing in human and conjunctival cells were followed by a 28-day short-term stability study under various conditions. The formulations were isotonic (280–300 mOsm/L), had a pH close to neutral (7–8), medium surface-tension values (<56 MN/m^−1^), and low (≈1 mPa·s) and medium (≈5 mPa·s) viscosities (compounded normal saline solution and artificial tear-based preparation, respectively). These values remained stable for 28 days under refrigeration. Microbiological stability was also excellent. Insulin potency remained in the 90–110% range in the compounded formulations containing normal saline solution when stored at 2–8 °C for 28 days, while it decreased in those based on artificial tears. Although both formulations were well tolerated in vitro, the compounded insulin diluted in a normal saline solution exhibited better cell tolerance. Preliminary data in humans showed that insulin in saline solution was an effective and safe treatment for persistent corneal epithelial defects. Compounded insulin eye drops diluted in normal saline solution could, therefore, constitute an emergent therapy for the treatment of persistent corneal epithelial defects.

## 1. Introduction

The cornea and conjunctiva are the outermost layers of the eye. They are both coated by the precorneal tear film that isolates the eye from the external environment. The cornea is an avascular, transparent, and highly innervated tissue. It has two main functions. It acts as a mechanical barrier, and it accounts for 75% of the refractive power of the eye. The epithelium is its outermost layer, consisting of 5–6 layers of nonkeratinized squamous epithelial cells. These are highly interconnected by tight junctions that act as an airtight seal and play an important role in establishing and maintaining cell polarity and barrier function [1]. This layer is constantly regenerated (every 7–10 days) by differentiation and maturation of corneal limbus stem cells [2,3]. A healthy epithelium is essential to protecting the eye from infection and preventing structural damage to deeper tissues.

Persistent corneal epithelial defect (PED) occurs when the corneal mechanisms responsible for epithelialization fail. In addition to causing ocular discomfort and compromising vision, PEDs may have other consequences, such as infection, corneal scarring, corneal fusion, and perforation. Multiple factors are related to corneal surface epithelialization deficit. Impaired epithelial adhesion, limbal stem-cell deficiency, trauma, drugs, and infection, among others, are some of the factors related to this problem [4].

Although there have been remarkable advances in recent years, pharmacotherapeutic agents capable of promoting corneal healing are still scarce [5]. The current gold-standard treatment for PEDs starts with conservative management and, for refractory cases, progresses to medical and surgical treatment. The first line of treatment is to halt all medication potentially toxic to the epithelium and to intensify eye lubrication by prescribing artificial tears (preservative free), prophylactic topical antibiotics, bandage soft-contact lenses, and punctal plugs [6]. The second line of treatment is to administer autologous serum eye drops and other blood products such as platelet-rich plasma eye drops [7,8]. If none of these options prove effective, more aggressive treatment, such as amniotic membrane transplantation (AMT) or surgery, would be necessary.

Topical insulin eye drops have emerged as a promising alternative PED treatment [9,10,11,12], as they offer a means of accelerating corneal re-epithelialization in patients in whom standard therapy is ineffective. Insulin is a peptide closely related to the insulin-like growth factor (IGF) able to stimulate keratinocyte migration and is involved in wound repair. Although insulin receptors have been found on both the human corneal surface and the tear film, the mechanism by which insulin promotes regeneration of the epithelium is not fully understood [13,14,15]. However, the importance of glucose’s role is already known for corneal cells to have adequate functionality. Glucose uptake is insulin independent in the eye, and it is mediated by an active glucose transporter, GLUT1, that increases after wounding to provide the metabolic energy necessary for cell migration and proliferation [16,17].

In the years since the use of topical insulin to treat corneal ulcers was first described, several authors (including this research group) have published case series showing that compounded insulin eye drops are an effective and safe means of treating PED [10,11,18,19,20,21]. The safety of topically applying insulin to the eye was first demonstrated by Bartlett et al., who applied insulin eye drops in single-dose concentrations of up to 100 IU/mL to eight healthy eyes [22]. Nevertheless, insulin eye drops are not yet available commercially and require preparation by a licensed compounding pharmacy from a marketed insulin drug designed for subcutaneous administration. Insulin eye drops have been prepared at different concentrations to treat several disorders. In single-dose and multi-dose studies, for example, formulations at 100 IU/mL were shown to be safe and effective at lowering blood glucose levels in nondiabetic humans [20,22]. More recent studies tested the efficacy of compounded insulin eye drops containing concentrations of 1–100 IU/mL for treating PEDs caused by infectious, immune, or neurotrophic etiologies [18,23,24]. Insulin eye drops at 1 IU/mL, 25 IU/mL, and 100 IU/mL were used to treat corneal epithelial lesions after surgery in diabetic patients [19], while a 25 IU/mL concentration was employed in patients with recurrent epithelial corneal erosion [23]. Refractory neurotrophic keratopathy was also treated with topical insulin eye drops at 1 IU/mL in nondiabetic patients [24].

As using ophthalmic insulin to treat corneal alterations is relatively new [25], there are no guidelines regarding the preparation, solvent selection, storage, and stability of compounded insulin eye drops.

Compounding insulin eye drops places a high burden on hospital pharmacists and entails a financial cost for hospitals and patients where long-term use is required. Furthermore, as the stability of the product has not yet been determined, it is discarded after 15 days of use [10]. These clinical needs and the scant evidence available make further evaluation of compounded insulin eye drops essential [26,27].

This paper evaluates the influence of two solvents on the short-term stability of compounded insulin eye drops at 1 IU/mL. The solvents selected are commonly employed in the preparation of eye drops and comprise an artificial tear formulation containing polyethylene glycol (PEG) and polypropylene glycol (PG), among other excipients, and an isotonic solution of sodium chloride (normal saline solution, NSS). The decision to utilize normal saline solution (NSS) was based on its safety profile on the ocular surface, its widespread utilization in clinical practice, low cost, and ease of preparation, thus ensuring that the workload in the Pharmacy Service is not unnecessarily increased.

To this end, full physicochemical evaluation was performed, in vitro tolerance in cell cultures (human corneal and conjunctival cell lines) was analyzed, and microbiological assays and insulin quantitation in the proposed insulin formulations were conducted using high-performance liquid chromatography (HPLC) under different temperature conditions (refrigeration and at room temperature). All the properties were assayed in closed containers and under simulated in-use conditions. Finally, efficacy studies using a saline-solution-based formulation were performed in patients with PEDs. To the authors’ knowledge, this is the first time that such a comprehensive evaluation of these solvents’ impact on compounded insulin eye drop stability has been performed.

## 2. Materials and Methods

### 2.1. Materials

Human regular insulin was obtained from Actrapid^®^ 100 IU/mL (Novo Nordisk A/S, Bagsværd, Denmark) (where 1 mL is equivalent to 3.5 mg of human insulin). Commercial eye drops (Systane Ultra from Allergan) based on a PEG and PG combination were also employed. Sodium chloride was obtained from Merck (Merck KGaA, Darmstadt, Germany) and Bbraun (Bbraun Medical, Barcelona, Spain). Recombinant human insulin was obtained from SAFC^®^, (Merck KGaA, Darmstadt, Germany). 3-[4,5-Dimethylthiazole-2-yl]-2,5-diphenyltetrazolium bromide (MTT) was obtained from Merck Life Science (Madrid, Spain). HPLC-grade trifluoroacetic acid (TFA) was acquired from Sigma Aldrich Solutions (Merck KGaA, Darmstadt, Germany), while UHPC-grade acetonitrile was purchased from Panreac Applichem (Barcelona, Spain). Water was purified using a MilliQ^®^ filtration system (Millipore Corporation, Billerica, MA, USA).

### 2.2. Preparation of Insulin-Based Eye Drops

Insulin eye drops were prepared by the Pharmacy Service of the San Carlos Clinical Hospital in a vertical laminar flow hood located in a C-class clean room using an aseptic technique. Topically instilled insulin was prepared at 1 IU/mL as per previous studies [10,11,28] from a commercial solution of human insulin for subcutaneous injection (Actrapid 100 IU/mL; Novo Nordisk A/S, Bagsværd, Denmark).

Two batches of compounded insulin eye drops were prepared following similar procedures. The formulations were obtained by two consecutive dilutions of the stock solution (Actrapid^®^) using a sterile syringe, achieving a final insulin concentration of 1 IU/mL. In both cases, the first dilution was made with an isotonic sodium chloride solution (NaCl 0.9%) and was followed by a second dilution with either (i) an isotonic sodium chloride solution (NaCl 0.9%) to obtain the formulation named Inclor or (ii) artificial tears based on a PEG and PG combination to obtain the formulation named Inpeg. For both preparations, a final sterile filtration through a polyethersulfone medium (0.22 µm pore size) was performed. Finally, the formulations were packaged in 10 mL sterile amber glass multi-dose eye drop bottles (Guinama, Valencia, Spain).

Both preparations were stored in a refrigerator at 2–8 °C. The shelf life of the compounded insulin eye drops was estimated by the pharmacy service as per previous studies [11,29]. Hospital ophthalmologists prescribed the application of 1 topical insulin eye drop every 6 h to all patients. The patients were provided with 2 bottles of Insys or 4 bottles of Inclor per month and instructed by the pharmacist on the proper handling, administration, conservation, and shelf life of the insulin-based eye drops.

### 2.3. Stability Study Design

To better understand the insulin’s stability behavior, a short-term stability study was designed. For this, 4 storage conditions were evaluated. Under Condition 1, the eye drop bottles were stored upright in unopened containers at 2–8 °C for 15 days, while Condition 2 consisted of an in-use test in which 1 drop was administered every 6 h daily, and the open bottles were kept refrigerated. To reflect the circumstances in which the product is used after storage, the unopened bottles were kept refrigerated for 15 days (as in Condition 1) and then subjected to in-use testing, administering 1 drop every 6 h daily for 15 days as per Condition 2 (Condition 1 + 2). Finally, another set of bottles was kept at room temperature under in-use conditions for 7 days to determine insulin stability during real-world patient usage (Condition 4).

Analyses were performed at the start (D0) of the study for all the conditions and at the 7th (D1) and 15th (D2) days of storage for Condition 1 and Condition 2. For Condition 1 + 2, tests were performed on the 15th day of storage in unopened bottles (D2) and then on the 21st (D3) and 28th (D4) days of storage under in-use conditions, while samples were analyzed on the 7th (D1) day of storage at room temperature.

At each time point, samples were taken and subjected to visual inspection and pH, osmolarity, viscosity, surface tension, insulin quantification, in vitro tolerance in corneal and conjunctival cells, and microbiological content analysis.

The short-term study design is shown in Table 1. Stability studies were conducted on 3 batches of 3 units to obtain a minimum of 3 independent measurements.

### 2.4. Physicochemical Characterization of Insulin Preparations

#### 2.4.1. pH

The pH of the formulations was measured using a pH meter (Mettler GLP 222 Crison, Hach Lange, Barcelona, Spain) equipped with a microelectrode (InLab, Mettler, Madrid, Spain). The samples were measured in triplicate.

#### 2.4.2. Osmolarity

Osmolarity was measured by the freezing-point depression technique using a Fiske^®^ single-sample micro-osmometer (model 210, Advanced Instruments, Norwood, MA, USA). Prior to sample measurement, proper osmometer function was tested by measuring a 290 mOsm/L standard. The samples were measured in triplicate.

#### 2.4.3. Surface Tension

A K-11 (Kruss) tensiometer (Kruss, Hamburg, Germany) and the Wilhelmy plate method were employed to analyze the surface tension of the insulin formulations under each storage condition. MilliQ water was used for calibration (72.0 ± 1 MN/m) prior to sample measurement. Before analysis, each formulation was pre-warmed to 33 °C and equilibrated for 3 min.

#### 2.4.4. Rheological Studies

A parallel plate system (60 mm diameter and 0.6 mm gap) attached to a Discovery HR1 hybrid rheometer (TA Instruments, New Castle, DE, USA) was employed to measure the viscosity of the samples. The viscosity was measured by increasing shear rates from 0 to 1000 s^−1^ in 30 steps. The study was carried out at room temperature and in triplicate.

### 2.5. In Vitro Tolerance Studies

#### 2.5.1. Cell Cultures

Two different human cell lines were employed to assess the in vitro tolerance of the developed preparations. Immortalized human corneal epithelial cells (HCECs; Evercyte GmbH, Vienna, Austria) were maintained at 37 °C under 5% CO_2_ in a humid atmosphere (95%). The medium was changed every 48–72 h. HCECs were kept in an EpiLife^®^ cell-culture medium (Life Technologies, Madrid, Spain) with EDGS^®^ 1X (Life Technologies, Madrid, Spain) and penicillin–streptomycin 1% (Life Technologies, Madrid, Spain) as supplementation. Likewise, immortalized human conjunctival epithelial cells (HConEpiCs; Innoprot, Bizkaia, Spain) were maintained under the same temperature, humidity, and CO_2_ conditions, while the medium was changed every 48 h. HConEpiCs were cultured using the IM-Ocular Epithelial Cell Medium Kit (Innoprot, Bizkaia, Spain). To ensure correct cell attachment and maintenance, the flasks were coated with collagen I (1 mg/mL) (Innoprot, Bizkaia, Spain).

#### 2.5.2. Cell Viability in Human Corneal and Conjunctival Epithelial Cell Lines

Cell viability assays to test the insulin preparations were performed in HCECs and HConEpiCs. For the HCEC and HConEpiC cell viability tests, 20,000 cells/well and 30,000 cells/well were seeded, respectively, in 96-well plates and incubated overnight. Briefly, the cells were exposed to the preparations (Inclor and Inpeg; 100 µL per preparation) at the specified time points for different exposure times (15 min and 1 h) to simulate short- and long-term treatments, respectively. The cells were then incubated for 4 h with an MTT solution (0.33 mg/mL) made in cell-culture media. Next, the supernatants were removed, and DMSO (100 μL) was added to each well. Finally, to ensure formazan crystal solubilization, the plates were taken to the spectrophotometer, shaken for 5 min, and measured at 550 nm. Benzalkonium chloride (BAK) at 0.005% was employed as a positive control for cell toxicity [30]. Cells treated with the cell culture were considered a negative control and equivalent to 100% cell survival.

### 2.6. Insulin Quantitation

A simple and specific method for analyzing human insulin using RP-HPLC was used as per a previous study [31]. The equipment employed was a UHPLC (Acquity Arc Bio, Waters, Barcelona, Spain) coupled with data collection and processing software (Empoware 3, EMOMB01512 software support ID, Waters, Barcelona, Spain). The analyses were performed using an XBridge™ PREMIER peptide BEH C18 130 Å 2.5 µm 4.6 mm × 10 cm as a stationary phase (Waters, Barcelona, Spain). The mobile phase consisted of 1% TFA in water and acetonitrile (70:30, *v*/*v*) that was linearly changed to 60:40 (*v*/*v*) over 5 min and kept constant for 5 min to achieve the initial proportion of 70:30 (*v*/*v*). Mobile phases were filtered by 0.22 µm membrane and degassed prior to use. The flow rate was set at 1 mL/min, the injection volume at 20 µL, and the eluent was monitored at 214 nm. The samples were kept at room temperature during analysis and were injected in duplicate. The samples were measured in triplicate.

### 2.7. Microbiological Analysis

Microbiological stability was assessed in 3 separate batches of each of the two preparations under each storage condition. Three culture media were used for the microbiological study: blood agar (general culture medium for fastidious microorganisms, gram-positive and gram-negative bacteria, filamentous fungi, and yeasts); Sabouraud (media containing glucose are especially suitable for dermatophytes, while those containing maltose are to be preferred for yeasts and molds); and nutrient broth (general liquid medium that supports the growth of a wide range of nonfastidious organisms). The study was not carried out under anaerobic conditions, as this does not reflect real-world circumstances.

The samples were incubated for 4 days at 35 °C and then examined for the presence of microbial colonies. To determine whether any inhibitory or antimicrobial properties would prevent the sterility test from detecting the presence of viable microorganisms, a suitability test was carried out prior to the sterility test. Growth could be observed in both formulations, as described in the corresponding USP monograph [32].

### 2.8. Preliminary Clinical Efficacy Study

The Cornea Unit at San Carlos Clinical Hospital has been offering patients with PEDs off-label treatment with topical insulin since October 2019. Our research group has published two prior papers describing the efficacy and safety of 1 IU/mL insulin eye drops containing fast-acting insulin (Actrapid; Novo Nordisk A/S, Søborg, Denmark) in a solution designed for subcutaneous injection and diluted in a commercial PEG and PG-based artificial tear formulation [9,10]. This study presents a case series of patients with PED treated with the novel Inclor preparation every 6 h (4 times a day). In this study, 9 eyes of 9 patients were treated with Inclor. The study was conducted in accordance with the ethical guidelines of the Declaration of Helsinki and informed consent was obtained from all subjects involved in the study. As previously mentioned, the pharmacist instructed each patient on the proper handling and administration of the drops, as well as on proper conservation. Before starting treatment with topical insulin, the following patient variables were recorded: age, sex, previous ocular disease and surgery, PED etiology, time since diagnosis, concomitant treatment, visual acuity (VA), and epithelial defect area. After starting treatment, VA, epithelial defect area, topical treatment, need for AMT or other surgeries, and recurrence were evaluated in each visit. To evaluate the corneal epithelial defect area in anterior segment imaging, image analysis software (ImageJ, version number 1.53) was used. The rate (initial PED area divided by days till epithelialization in mm^2^/day) and the time until complete healing of the epithelial defect were considered efficacy endpoints.

### 2.9. Statistical Analysis

The two insulin preparations, Inclor and Inpeg, were prepared in separate batches (n = 3), and each batch was analyzed in triplicate. Data are presented as mean ± standard deviation. For cell-culture experiments, each sample was analyzed in 7 different wells on the same day (technical replicates). This procedure was repeated on 3 different days (biological replicates) to ensure experiment reproducibility. Regarding insulin quantitation, a variation of the concentration outside the limits ranging between 90 and 110% of the initial concentration was considered an unacceptable level of stability. The stabilities of the compounded formulations were evaluated by comparing the results at each point with the data from the beginning of the study to determine if significant differences in the means appeared. A two-sample *t*-test (Student’s *t*-test) was performed to determine the level of significance of the physicochemical and quantitation data obtained from the stability study. Reported *p*-values indicate the level of significance (ns; *: *p* ≤ 0.05; **: *p* ≤ 0.01; ***: *p* ≤ 0.001).

When significant differences in the variances appear (after a variance test), an approximation for the t* distribution is made which involves Student’s t distribution having degrees of freedom approximated by quantity [33]. StatGraphics Centurion 19 software (19.3.03-64 bit version number) was used for the statistical analysis.

## 3. Results

### 3.1. Characterization of Insulin Eye Drops

The two compounded formulations (Inclor, diluted in NSS, and Inpeg, diluted in PEG and PG-based artificial tears) containing insulin at 1 IU/mL were physicochemically characterized immediately after preparation (D0) (Table 2).

Both formulations were transparent, and no visible foreign particles were detected. Inclor had a neutral pH, while Inpeg had a basic pH. Surface-tension values were between 51 and 55 mN·m^−1^ for both solutions and were lower than the reference sample (water value: 72.0 ± 1 mN/m). The viscosity of the Inpeg increased more than 4.5-fold mPa versus that of the Inclor. In both preparations, osmolarity was in the physiological range of healthy tears (≈300 mOsm/L) [34].

### 3.2. Physicochemical Stability of Insulin Eye Drops

During the experimental short-term stability period, all samples remained limpid, uncolored, and transparent under all examined conditions. No visible foreign particles were found.

Throughout the study period, the pH did not change versus the initial values (7.00 for Inclor and 8.12 for Inpeg) by more than 0.15 and 0.25 units, respectively.

Regarding surface tension, the difference among the various storage conditions and time sets evaluated was less than 4% (2 mN/m) for Inclor and less than 11% (6 mN/m) for Inpeg. Viscosity did not vary by more than 0.02 and 0.52 units for Inclor and Inpeg, respectively, during the stability period studied. As regards osmolarity, in the case of Inclor, differences with the initial value (290 mOs/L) were not more than 4% (12 mOsm/L) throughout the study. Inpeg meanwhile showed a difference with the initial mean osmolarity value (281 mOsm/L) of less than 6% (17 mOsm/L) during the stability period.

Despite the differences observed in certain parameters and storage conditions, these variations in physicochemical properties are not expected to have a clinically relevant effect.

Table 3 and Table 4 show data regarding physicochemical stability for Inclor and Inpeg, respectively.

### 3.3. Cell Viability Studies

To evaluate the in vitro tolerance and suitability of the compounded formulations, the toxicity of Inclor and Inpeg was assessed in HCECs and HConEpiCs after 15 min and 1 h exposures to each preparation. Tests to establish the shelf life of the formulations under the short-term stability study were also performed.

Freshly prepared Inclor formulations were well tolerated in corneal and conjunctival cells at both exposure times, presenting survival values close to 100%. Inclor maintained optimal tolerance, recording corneal and conjunctival viability values above 90% under the four storage conditions (Figure 1).

The compounded Inpeg formulations showed acceptable conjunctival cell tolerance at both exposure times on the days and under the conditions evaluated. In vitro tolerance was always above 84% (Figure 2), meeting the previously established acceptance criteria (cell viability > 80%) for the development of topical ophthalmic formulations [30,35].

Inpeg’s corneal viability was lower at 70–95% (Figure 2). Cell viability was lower at 1 h corneal exposures than at 15 min exposures under each storage condition, exhibiting a decreasing trend.

### 3.4. Insulin Quantitation in the Preparations

The theoretical initial insulin concentration in the two compounded preparations, Inpeg and Inclor, was 1 IU/mL, and the experimental values at the beginning of the study do not differ significantly from one (*p* = 0.574 for Inclor and *p* = 0.104 for Inpeg).

The insulin concentration in the Inclor preparations remained constant and did not show any significant differences between the concentration tested at the beginning of the study and those obtained throughout it (*p* > 0.05) at refrigerator temperature. The insulin concentration was close to the theoretical concentration of 1 IU/mL under Condition 1 (15 days in unopened bottles), Condition 2 (in-use test for 15 days), and Condition 1 + 2 (in-use test for 15 days after 15 days of storage in unopened bottles) when stored under refrigeration. However, a significant decrease was observed for the amount of insulin after 7 and 15 days when the preparation was stored at 25 °C (*p* = 0.0098 and *p* = 0.0107 after 7 and 15 days of storage, respectively), presenting values slightly above 70% of the initial concentration (Figure 3A). When stored under refrigeration, the insulin concentration remained stable within the 90–110% range (at the end of storage, the insulin concentrations were 1.03 ± 0.01 µg/mL, 1.002 ± 0.043 µg/mL, and 1.032 ± 0.066 µg/mL, respectively, for Conditions 1, 2, and 1 + 2), but the concentrations decreased significantly when stored at room temperature (0.759 ± 0.051 µg/mL).

Regarding Inpeg, the amount of insulin was significantly lower than the initial concentration at all time points (*p* < 0.05), except when stored in closed containers under refrigeration (Condition 1); albeit in this case, storage was only for 7 days (*p* = 0.831). Considering the best case (Condition 1), in which unopened bottles were stored under refrigeration, the insulin concentration decreased from 1.13 ± 0.11 IU/mL (mean ± standard deviation) after 1 week (*p* = 0.831) to 0.932 ± 0.02 IU/mL after 2 weeks (*p* = 0.013). Meanwhile, under Condition 2 (in-use test for 15 days with bottles stored at 2–8 °C), the insulin concentration significantly diminished to 0.89 ± 0.11 IU/mL and 0.85 ± 0.02 IU/mL after 1 and 2 weeks of storage, respectively. In the formulations stored under Condition 1 + 2, the amount of insulin decreased significantly after 15 days of storage in closed containers under refrigeration (0.85 ± 0.02 IU/mL and *p* = 0.003 mPa versus freshly prepared formulations); the insulin then remained stable (*p* > 0.05) after 7 and 15 days of in-use testing (0.86 ± 0.09 IU/mL and 0.88 ± 0.03 IU/mL) (Figure 3B).

### 3.5. Sterility Assay

None of the analyzed preparations in unopened (Condition 1), opened (Condition 2 and Condition 3), and both unopened and opened (Condition 1 + 2) multi-dose eyedroppers showed any evidence of microbial growth when incubated for at least 4 days at 35 °C in the culture medium assayed during the entire stability study. The compounded insulin eye drop sterility was not affected by either the storage temperature or the storage condition evaluated.

### 3.6. Clinical Study

A preliminary clinical study with nine patients treated with Inclor (five males and four females; mean age 55.1 ± 23.9 years) was conducted. The PED etiologies were as follows: neurotrophic (six patients, 67%), immune-mediated (two patients, 22%), and chronic alterations of the ocular surface (one patient, 11%). The mean time between diagnosis and the start of administration of compounded insulin eye drops was 14.8 ± 5.4 days, and the mean epithelial defect area at the beginning of treatment was 8.2 ± 5.8 mm^2^. All patients achieved PED epithelialization in a mean time of 16.6 ± 10.8 days. The epithelialization rate was 0.9 ± 1.0 mm^2^/day. Topical ocular insulin was well tolerated and no adverse events (including infection) were reported with the treatment. Only one patient experienced PED recurrence and none required AMT or further surgery. Table 5 summarizes patient demographics, baseline characteristics, time until re-epithelialization, and re-epithelialization rate (mm^2^/day). As an example, Figure 4 shows the evolution of Patient 9 with treatment with Inclor.

## 4. Discussion

Insulin has been shown to be vital for achieving efficient repair of corneal epithelial defects when administered by a systemic and ocular topical route [24,36,37]. From a clinical point of view, treating PEDs with topical insulin is advantageous because direct application to the injured eye is faster and more effective than other options. Moreover, the efficacy of treating PEDs with topical insulin has been demonstrated clinically [9,10,24]. This recent discovery has led to an increase in hospital pharmacy services compounding ocular topical insulin formulations, since no commercial insulin eye drops designed to repair damage to the corneal epithelium are currently available. These preparations are compounded by pharmacists under aseptic conditions using marketed insulin designed for intramuscular injection, e.g., Actrapid^®^ 100 IU/mL (Novo Nordisk, Bagsværd, Denmark). The procedure for preparing compounded insulin eye drops depends on the hospital in which the topical formulations are made. There are differences in insulin potency (from 1 to 100 IU/mL) and in the solvents used as the vehicle (artificial tears based on PEG and PG, balanced salt solution, or NSS). Various concentrations of topical insulin (25 IU/mL, 50 IU/mL, and 100 IU/mL, supplied by Actrapid^®^) were obtained after dilution with NSS to treat corneal epithelial defects in diabetic patients after corneal debridement during intraocular surgery. This study found that the most effective insulin concentration was 25 IU/mL and that higher insulin concentrations were related to slower cell migration during the healing process [18]. Notwithstanding this, diabetic patients with postoperative corneal epithelial defect after vitreoretinal surgery—which complied with the clinical criteria—received compounded insulin eye drops at a higher concentration, 50 IU/mL (supplied by Actrapid^®^), likewise diluted in NSS. In this study, the patients administered one drop every 6 h (four times daily) of formulations at 1 IU/mL until the epithelial defect healed without appreciable local or systemic side effects [21]. A concentration of 1 IU/mL of insulin in the compounded ophthalmic formulations was selected based on previous studies of the research group [9,10] that showed this concentration as effective and safe for treating PEDs. Currently, there is no demonstrated either minimum or maximum insulin-effective concentration in compounded ophthalmic drops for PED, though we consider this concentration, 1 IU/mL, as a precautionary approach to minimize drug exposure and any, albeit low, potential toxicity.

According to the technical data sheet, Actrapid^®^ consists of 100 IU of recombinant insulin and trace excipients, such as zinc chloride, glycerol, m-cresol, sodium hydroxide, hydrochloric acid, and water for injection. While zinc’s contribution to skin-healing processes has been demonstrated [38], some research has reported that regular insulin has superior skin-healing properties to a zinc solution and has observed a synergistic effect between insulin and zinc [39]. The presence of trace amounts of zinc in the compounded insulin eye drops may have a synergistic effect with insulin in terms of healing. However, further studies evaluating this relationship are required. Another substance found in Actrapid^®^ is m-cresol, which exhibits antimicrobial properties and gives insulin stability in an aqueous solution [40]. This molecule, together with the sterile preparation process and final sterilization of the formulations, may have contributed to keeping the formulations free of microbial contamination. However, m-cresol also triggers undesirable side effects, such as hypersensitivity reactions in a dose- and concentration-dependent manner. It is already known that m-cresol produces severe corneal damage when administered to the ocular surface at 5%, while at 1% it does not cause any corneal alteration [41]. The theoretical m-cresol concentration in the marketed insulin drug used in the compounded insulin eye drops is 0.03%, which may explain the acceptable tolerance values reported in corneal and conjunctival cells in this paper, especially for Inclor.

There are no common guidelines for preparing insulin eye drops, nor are there any comprehensive studies endorsing either a minimum effective insulin dosage or an absolute compounded formulation stability. In fact, due to the high burden on hospital pharmacy services and the immediacy of this finding in humans, to date, there is scant information on the stability of these compounding insulin preparations. Thus, the storage conditions and shelf life are determined by the marketed formulations intended for other uses and formulated with different insulin potencies. The differences in the solvents employed in the compounded eye drops may affect the physicochemical stability of the insulin. This study evaluated two different solvents—an isotonic saline solution (Inclor), and artificial tears based on a PEG and PG mixture (Inpeg)—by performing physicochemical, in vitro tolerance, microbiological, and quantitation studies followed by efficacy evaluation in PED patients. Patients received four bottles of Inclor or two bottles of Inpeg. Since Inclor contains NSS, it is preservative free. According to the general consensus regarding ophthalmic preparations, the estimated shelf life is 7 days. Conversely, the Inpeg formulations contained the preservative polyquaternium-1, which was employed in the artificial tears as the solvent. The pharmacy service determined these formulations’ estimated shelf life at 14 days [42]. Regarding the physicochemical properties of the freshly made preparations, Inclor had an almost neutral pH (7.00 ± 0.10) while Inpeg had a basic pH (8.12 ± 0.01). The pH value of Inclor is related to the composition of the Actrapid^®^, which contains NaOH and HCl to give it a neutral pH and, therefore, make it suitable for intramuscular injection. Inpeg, on the other hand, had a basic pH similar to the value of the artificial tears used as the vehicle. Both preparations’ surface-tension values were slightly higher (51–55 MN/m) than the ones observed in the precorneal tear film (40–46 mN·m^−1^), though the two preparations’ extensibility on the ocular surface can be considered acceptable, given that PED treatment is a short-term therapy. If long-term treatment were required (e.g., for dry-eye disease), it would be beneficial to consider reformulating the two preparations based on the physiological range of the surface-tension values [43]. The viscosity of Inpeg was 4.5 times greater than that of Inclor due to the composition of the vehicle, which contains PEG, PG, and other polymers such as hydroxypropyl guar that increase the intrinsic viscosity of the compounded preparation. Inpeg’s augmented viscosity could be beneficial, as it increases the insulin retention time on the ocular surface, thus producing a prolonged effect [44].

The viability of the compounded formulations was studied in corneal (HCEC) and conjunctival (HConEpiC) cells. The studies revealed that Inclor was well tolerated in corneal and conjunctival cells after both short (15 min) and long (1 h) exposure times. The Inpeg tolerance was good in conjunctival cells and slightly poorer in corneal cells, an effect that may be due to corneal cells being more sensitive to the vehicle components in the Inpeg preparation. Artificial tears employed as a solvent in Inpeg eye drops contain components that, when combined with certain ingredients found in Actrapid^®^—such as polyquad^®^ (polyquaternium-1 at 0.001%), which is used as a preservative—may produce greater reactivity in corneal cells. It is well known that preservatives in eye drops can induce histopathological, inflammatory, and toxic changes on the ocular surface. Preservatives such as benzalkonium chloride (BAK) and polyquaternium-1 are employed to keep ophthalmic formulations free of microorganisms. BAK is the most widely employed preservative, although its use in ophthalmology is decreasing since BAK produces ocular surface alterations [45,46]. Withdrawing BAK from eye drops improves the tolerance of ophthalmic formulations, especially in the case of therapies for chronic diseases such as glaucoma [47]. Meanwhile, polyquaternium-1 has been shown to produce corneal epithelial damage [48], in laboratory, experimental, and clinical studies, although concentrations below 0.5% are considered safe for the eye [49]. No microbiological contamination was observed in the preparations throughout the short-term stability study. Therefore, further studies evaluating each ingredient in combination with those in Actrapid^®^ are needed to determine this relationship.

Quantitation by HPLC found that insulin remained unchanged in both preparations when freshly made. Additionally, the preparation procedure avoids microbiological contamination.

This study also assessed the extent to which the temperature and conditions under which each selected vehicle—isotonic saline solution and PEG and PG-based artificial tears—are stored affect its physicochemical properties, insulin stability, in vitro tolerance, and microbiological contamination. The findings of this research indicate that, for both compounded preparations, all parameters favored physicochemical stability after both 1 month and 1 week of storage at 2–8 °C (Condition 1, 2, and 1 + 2) and 25 °C (Condition 3), respectively. Visual appearance, color, turbidity, and pH all remained unchanged throughout the study, as did the surface tension, viscosity, and osmolarity of the insulin eye drops. Cell tolerance remained at acceptable values, and no microbiological contamination was observed.

However, insulin content stability is not only affected by storage conditions; it is also affected by the type of solvent employed. Compounded insulin (1 IU/mL) eye drops in NSS (Inclor) were stable for up to 28 days when refrigerated, while when an artificial tears-based solvent was used (Inpeg), the insulin content decreased significantly over time under refrigeration. At room temperature, the insulin potency decreased significantly in both compounded preparations regardless of the solvent used. These findings are at variance with those of Cuartero-Martínez et al., who observed that insulin eye drops at 25 IU/mL in NSS and packaged in amber glass containers remained stable for 120 days when refrigerated, frozen, or stored at room temperature and protected from light. When a balanced salt solution was used as the solvent, the authors found that the stability decreased to 90 days when the formulations were frozen but remained at 120 days at room temperature and under refrigeration [27]. These differences can be explained by insulin potency and by the interactions between the salts and the ingredients in the reference product employed in the compounded insulin eye drops and require further study to determine the cause.

Dissimilarity in insulin stability is also found when an in-use test is studied. While compounded insulin eye drops containing NSS (Inclor) maintained insulin content, the insulin in Inpeg decreased significantly. Meanwhile, a recent study by Le Nguyen et al. evaluated the physicochemical and microbiological stability of compounded insulin (1 IU/mL) eye drops using artificial tears with a PG and PEG base as a vehicle and insulin lispro as an active substance. The eye drops were packaged in low-density polyethylene (LDPE) multi-dose eye droppers [26]. These authors found that, while the insulin remained stable for 12 months in unopened eye droppers stored at 4 °C, in simulated use and stored at 4 °C, it only remained stable for 1 month. These findings differ from those of our research group, in which insulin in the solvent based on artificial tears (PG and PEG mixture) decreased in potency after 7 days of storage at 4 °C but remained in the 90–110% range when NSS was used as the solvent. This circumstance may be due to the differences between the insulin employed in the marketed drug and the conditioning materials employed in the aforementioned study and this one (regular insulin from Actrapid^®^ and ophthalmic-grade glass as conditioning material). Insulin stability and the conditioning material employed are currently a source of controversy in the field. Adsorption is a reversible process that can occur with both plastics and glass and is greater at low insulin concentrations [50]. Although earlier studies indicate that glass adsorbs insulin in high proportions [51], more recent research shows that insulin adsorption tends to occur more with ethylene–vinyl acetate plastic than with glass [52]. Insulin adsorption in plastic materials has been evaluated, and it was concluded that polypropylene was the material that produced the least insulin adsorption, followed by polyethylene or polyvinyl chloride [53,54,55]. The most conservative option for packaging insulin eye drops would be to use glass containers with a high-density polyethylene dropper, simulating the packaging material of Actrapid^®^ vials. It follows that further studies are needed to evaluate the interactions between insulin and the packaging materials and between insulin and each of the components of the compounded insulin eye drops.

Preliminary clinical results for patients treated with Inclor were promising in terms of efficacy and safety. In the present case series, all patients achieved epithelialization in a mean time of 16.6 ± 10.8 days and presented an epithelialization rate of 0.9 ± 1.0 mm^2^/day. In a larger case series of patients treated with Inpeg [9], epithelialization was achieved in 51 patients (84%), and the mean time to re-epithelialization was 32.6 ± 28.3 days (epithelialization rate: 0.51 ± 0.55 mm^2^/day). Recurrence rates seem to be similar, as 11% recurrence was observed with both solvents. Inclor and Inpeg are well tolerated, and no adverse events have been reported with any of the preparations. This clinical study is considered preliminary, as it is the first time reporting the use of insulin in NSS compounded ophthalmic drops. This study was designed as a parallel trial in which each patient received one treatment (Inclor or Inpeg). A limitation of this study is the absence of a control group and a wash-out period. Considering the clinical success achieved with Inpeg—using artificial tears based on PG and PEG as the solvent—and its insulin stability, more studies on the minimum effective insulin potency for PED treatment are required. In addition, these findings suggest that both preparations (Inclor and Inpeg) and, consequently, both solvents (NSS and artificial tears containing PEG and PG, among other excipients) are suitable for PED therapy, even though the patients included in the case series of the latter had larger PEDs, and the results may not be completely comparable. Nevertheless, the clinical results for Inclor are also excellent and suggest it is at least as good as Inpeg.

## 5. Conclusions

Topical compounded insulin eye drops promote corneal re-epithelialization and are considered a promising strategy to cure persistent corneal defects. There are not currently any guidelines regarding the most effective dose nor are there any standardized preparation procedures for insulin-based compounded eye drops, though a 1 IU/mL concentration has been employed frequently in marketed subcutaneous insulin. In addition, insulin stability depends on the insulin employed, the packaging, and the storage conditions. Each ingredient in the compounded preparation, including preservatives, poly-alcohols, polymers, salts, and electrolytes, can influence the stability of the final formulation.

This study demonstrates that the stability of ocular topical insulin at 1 IU/mL depends on the composition of the solvent in which it is prepared and stored. NSS has been shown to maintain insulin stability for up to 28 days; in open bottles simulating in-use conditions, it was maintained for 15 days after 15 days of storage in closed containers at 4 °C. Moreover, insulin-based preparations in NSS have been shown to be effective and nontoxic at 1 IU/mL for the treatment of PEDs. Another advantage of this preparation is the absence of the interactions that might occur when more complex solvents are used, in addition to the simplicity and low cost of NSS.

Thus, the findings of this study support the need for a deeper understanding of the interactions between solvent excipients and insulin. They also support the need to optimize a formulation based on topical insulin as an emergent therapy for treating PEDs.

## Figures and Tables

**Figure 1 pharmaceutics-16-00580-f001:**
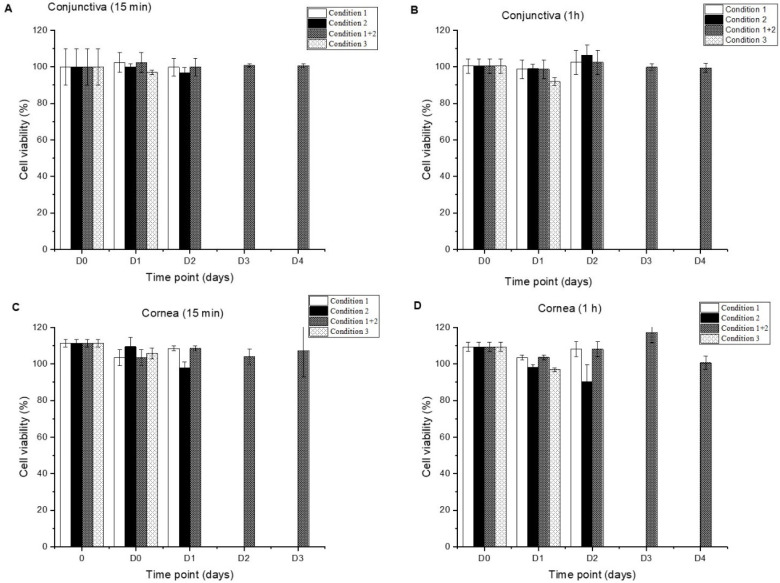
Cell viability of Inclor after 15 min (**A**) and 1 h (**B**) exposures in HConEpiCs and after 15 min (**C**) and 1 h (**D**) exposures in HCECs during the short-term stability study. Storage Condition 1 is 15 days in unopened bottles under refrigeration; Storage Condition 2 is in-use test for 15 days under refrigeration; Storage Condition 3 is 7 days at room temperature; and Storage Condition 1 + 2 is in-use test for 15 days after 15 days of storage in unopened bottles under refrigeration.

**Figure 2 pharmaceutics-16-00580-f002:**
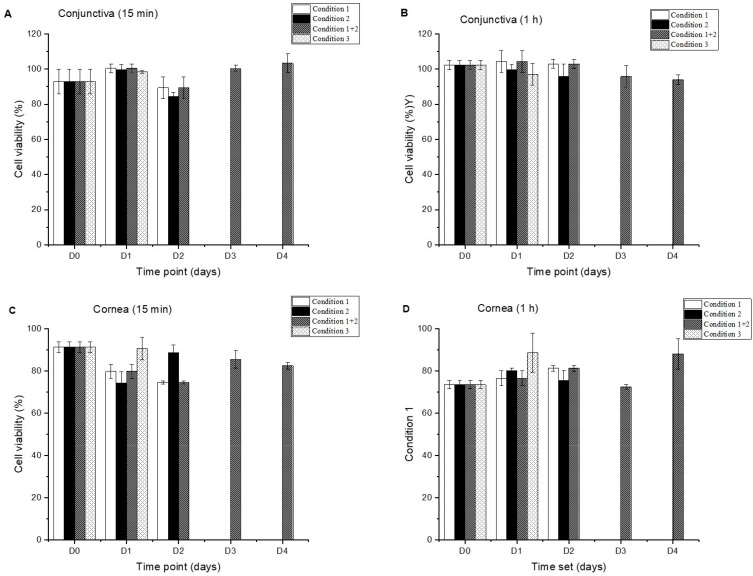
Cell viability of Inpeg after 15 min (**A**) and 1 h (**B**) exposures in HConEpiCs and after 15 min (**C**) and 1 h (**D**) exposures in HCECs during the short-term stability study.

**Figure 3 pharmaceutics-16-00580-f003:**
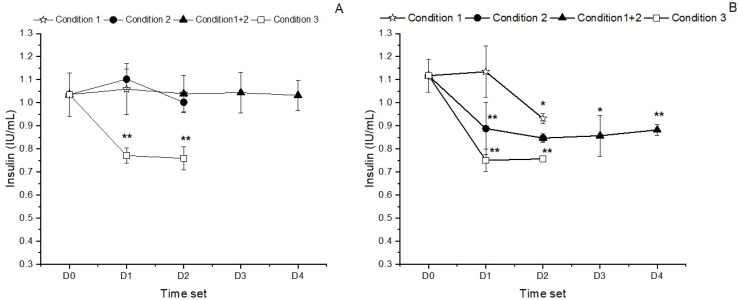
Insulin quantitation (%) for compounded Inclor (**A**) and Inpeg (**B**) throughout the stability study. Storage Condition 1 is 15 days in unopened bottles under refrigeration; Storage Condition 2 is in-use test for 15 days under refrigeration; Storage Condition 3 is 7 days at room temperature; and Storage Condition 1 + 2 is in-use test for 15 days after 15 days of storage in unopened bottles under refrigeration. * *p* ≤ 0.05; ** *p* ≤ 0.01.

**Figure 4 pharmaceutics-16-00580-f004:**
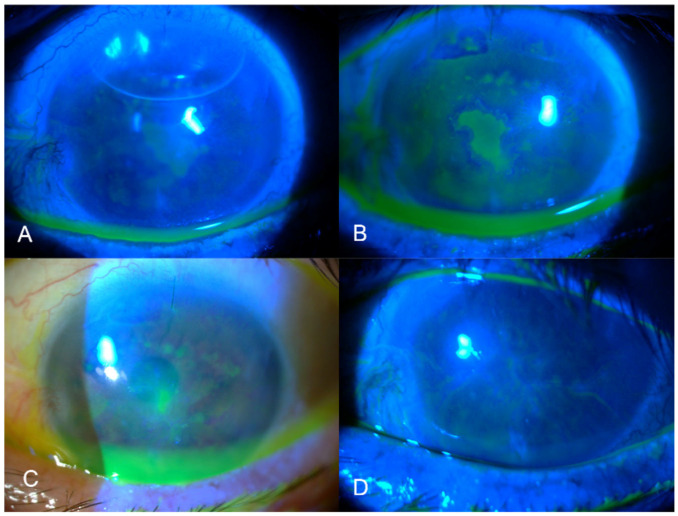
Serial slit-lamp images of a neurotrophic persistent epithelial defect treated with compounded insulin eye drops in isotonic saline solution (Inclor), Patient 9. (**A**) Baseline; (**B**) 4 days after initiating topical insulin treatment; (**C**) after 15 days’ treatment; and (**D**) after 22 days, when the epithelium healed.

**Table 1 pharmaceutics-16-00580-t001:** Overview of the short-term stability study design.

Condition	Storage Conditions	Time Point (days)
1	2–8 °C unopened container	D0, D1, D2 **
2	2–8 °C in-use conditions *	D0, D1, D2 **
1 + 2	2–8 °C unopened container followed by a 15-day period of in-use conditions	D0, D2, D3, D4 **
3	25 °C in-use conditions *	D0, D1 **

* In-use conditions indicate that containers are open, and a drop is released every 6 h. ** D0 = day 0 (freshly prepared formulations, start of study); D1 = day 7; D2 = day 15; D3 = day 21; D4 = day 28.

**Table 2 pharmaceutics-16-00580-t002:** Physicochemical characterization of compounded insulin eye drops after preparation (D0).

Formulation	Appearance	pH	Surface Tension (mN·m^−1^)	Viscosity (mPaMPa·s)	Osmolarity (mOsm/L)
Inclor	Transparent solution with no foreign particles in suspension	7.00 ± 0.10	55.17 ± 0.52	1.00 ± 0.01	289.93 ± 14.68
Inpeg	Transparent solution with no foreign particles in suspension	8.12 ± 0.01	50.80 ± 1.39	4.79 ± 0.05	281.27 ± 4.42

D0 corresponds to the start of the short-term stability study.

**Table 3 pharmaceutics-16-00580-t003:** Physicochemical stability of Inclor under each storage condition.

Parameter		Storage Condition			
	Time Point	1	2	3	1 + 2
pH	D0	7.00 ± 0.10	7.00 ± 0.10	7.00 ± 0.10	7.00 ± 0.10
	D1	7.05 ± 0.04	7.07 ± 0.03	7.25 ± 0.10 *	7.05 ± 0.04
	D2	7.14 ± 0.03	7.04 ± 0.03	-	7.14 ± 0.03
	D3	-	-	-	7.15 ± 0.04
	D4	-	-	-	7.01 ± 0.05
Surface Tension (mN·m^−1^)	D0	55.17 ± 0.52	55.17 ± 0.52	55.17 ± 0.52	55.17 ± 0.52
	D1	54.78 ± 1.64	54.13 ± 0.29	53.10 ± 0.89 *	54.78 ± 1.64
	D2	54.13 ± 1.86	53.64 ± 0.49 *	-	54.13 ± 1.86
	D3	-	-	-	53.83 ± 0.71
	D4	-	-	-	52.97 ± 0.43
Viscosity (mPaMPa·s)	D0	1.00 ± 0.01	1.00 ± 0.01	1.00 ± 0.01	1.00 ± 0.01
	D1	0.98 ± 0.03	1.00 ± 0.00	1.01 ± 0.01	1.00 ± 0.01
	D2	0.99 ± 0.00	1.01 ± 0.05	-	1.00 ± 0.01
	D3	-	-	-	1.04 ± 0.05
	D4	-	-	-	1.02 ± 0.01
Osmolarity (mOsm/L)	D0	289.93 ± 14.68	289.93 ± 14.68	289.93 ± 14.68	289.93 ± 14.68
	D1	283.56 ± 1.17	291.56 ± 4.43	286.00 ± 1.76	283.56 ± 1.17
	D2	285.53 ± 5.55	290.02 ± 6.06	-	285.53 ± 5.55
	D3	-	-	-	283.87 ± 1.98
	D4	-	-	-	295.69 ± 7.92

* *p* ≤ 0.05. The empty cells indicate the absence of collected data according to the stability study design. Storage Condition 1 is 15 days in unopened bottles under refrigeration; Storage Condition 2 is in-use test for 15 days under refrigeration; Storage Condition 3 is 7 days at room temperature; and Storage Condition 1 + 2 is in-use test for 15 days after 15 days of storage in unopened bottles under refrigeration.

**Table 4 pharmaceutics-16-00580-t004:** Physicochemical stability of Inpeg under each storage condition.

Parameter		Storage Condition			
	Time Point	1	2	3	1 + 2
pH	D0	8.12 ± 0.01	8.12 ± 0.01	8.12 ± 0.01	8.12 ± 0.01
	D1	7.92 ± 0.01 ***	7.93 ± 0.00 ***	8.13 ± 0.01	7.92 ± 0.01
	D2	7.88 ± 0.01 ***	7.88 ± 0.00 ***	-	7.88 ± 0.01
	D3	-	-	-	8.13 ± 0.01
	D4	-	-	-	8.00 ± 0.00
Surface Tension (mN·m^−1^)	D0	50.80 ± 1.39	50.80 ± 1.39	50.80 ± 1.39	50.80 ± 1.39
	D1	53.87 ± 0.25 *	53.63 ± 0.57	51.63 ± 3.79	53.87 ± 0.25
	D2	55.09 ± 0.41 **	53.87 ± 0.55	-	55.09 ± 0.41
	D3	-	-	-	49.14 ± 0.27
	D4	-	-	-	49.70 ± 0.82
Viscosity (mPaMPa·s)	D0	4.79 ± 0.05	4.79 ± 0.05	4.79 ± 0.05	4.79 ± 0.05
	D1	4.53 ± 0.37	4.89 ± 0.02	4.81 ± 0.34	4.53 ± 0.37
	D2	4.74 ± 0.06	4.95 ± 0.15	-	4.74 ± 0.06
	D3	-	-	-	4.98 ± 0.02
	D4	-	-	-	5.05 ± 0.08 **
Osmolarity (mOsm/L)	D0	281.27 ± 4.42	281.27 ± 4.42	281.27 ± 4.42	281.27 ± 4.42
	D1	285.82 ± 3.91	291.78 ± 3.71 *	282.22 ± 2.52 *	285.82 ± 3.91
	D2	288.33 ± 0.67	286.11 ± 2.12	-	288.33 ± 0.67
	D3	-	-	-	289.22 ± 7.96
	D4	-	-	-	298.18 ± 6.04 **

* *p* ≤ 0.05; ** *p* ≤ 0.01; *** *p* ≤ 0.001. The empty cells indicate the absence of collected data according to the stability study design.

**Table 5 pharmaceutics-16-00580-t005:** Clinical characteristics and outcomes of patients treated with compounded insulin eye drops using an isotonic saline solution as a vehicle (Inclor).

Patient	Age (Years)	Sex	Ocular Diseases	Etiology	Time Since Diagnosis (Days)	Visual Acuity	PED Area (mm^2^)	Previous Treatment	Re-Epithelialization (Days)	AMT	Surgery	Recurrence	Re-Epithelialization Rate (mm^2^/Day)
1	51	Male	Uveitis, glaucoma, retinal detachment	Neurotrophic	16	0.4	11.04	Tear substitutes, erythromycin ointment	19	0	0	0	0.58
2	71	Female	Pemphigoid	Immune-mediated	11	0.3	6.19	Tear substitutes, cyclosporine, autologous serum, doxycycline, prednisone	12	0	0	1	0.52
3	35	Male	Pemphigoid	Immune-mediated	27	0.2	3.74	Tear substitutes, fluormetholone, doxycycline	14	0	0	0	0.27
4	29	Female	Dry-eye disease	Neurotrophic	17	0.1	12.35	Tear substitutes, ciclosporin, autologous serum	22	0	0	0	0.56
5	50	Male	Glaucoma, ocular trauma	Neurotrophic	10	LP	0.54	Tear substitutes, erythromycin ointment, autologous serum, fluormetholone, moxifloxacin	13	0	0	0	0.04
6	31	Male	Congenital ptosis	Neurotrophic	12	0.6	8.26	Tear substitutes, erythromycin ointment	3	0	0	0	2.75
7	95	Female	Glaucoma, bullous keratopathy	Chronic alterations of the ocular surface	17	HM	11.67	Tear substitutes, bandage contact lens, ofloxacin, netilmicin	5	0	0	0	2.33
8	48	Female	Pterigium	Neurotrophic	13	0.5	1.59	Tear substitutes, tobramycin	39	0	0	0	0.04
9	86	Male	Uveitis, herpetic keratitis, glaucoma	Neurotrophic	10	0.05	18.67	Tear substitutes, tobramycin, ofloxacin	22	0	0	0	0.85

PED: persistent epithelial defect; AMT: amniotic membrane transplantation; LP: light perception; HM: hand movement.

## Data Availability

Data are contained within the article.

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
