# Peer review of "Topical Insulin Eye Drops: Stability and Safety of Two Compounded Formulations for Treating Persistent Corneal Epithelial Defects"

_pharmaceutics, 2024, doi:10.3390/pharmaceutics16050580_

Round 1
Reviewer 1 Report
Comments and Suggestions for Authors
This is a fascinating description of the use of insulin containing drops to treat defects of the corneal epitheium. The background must be amplified in terms of references to corneal glucose metabolism. It is also not clear why the authors chose an insuin concentrton of 1 IU/ml. The effects of insulin above a concentration maximum are stable. Do the authors have a dose response curve to show a relatinship between insulin concentration and
tear glucose concentration?
There is no control group for the clinical studies. It woud certainly be of value to have a comparison study using insuin drops in one cohort and perhaps a standard antibiotic drop in the other.
Certainly no reason to hold up pubication, but the above issues should be mentioned as incentive to further studies.
Author Response
We would like to extend our sincere gratitude to you for dedicating your time and expertise to review our manuscript titled 'Topical Insulin Eye Drops: Stability and Safety of Two Compounded Formulations for Treating Persistent Corneal Epithelial Defects' submitted to Pharmaceutics MDPI. Your insightful comments and constructive feedback have been invaluable in improving the quality and clarity of our work. Please, find enclose the point-by-pint responses to your comments and suggestions.
Thank you once again for your invaluable contribution to our research.

Reviewer 2 Report
Comments and Suggestions for Authors
In this study the authors analysed the impact of two different solvents on the stability of insulin eye drops at 1 IU/ml and found normal saline solution to be superior in providing longer stability during storage in addition to its efficacy, lower cost, and nontoxicity. Overall the study is thorough, well carried out and presented concisely. I applaud the authors for their work and its high clinical relevance ophthalmic practice. I'd only recommend to shorten the length of introduction and discussion.
Author Response
Dear Reviewer,
We want to express our appreciation for the time and effort you invested in reviewing our research manuscript, "Topical Insulin Eye Drops: Stability and Safety of Two Compounded Formulations for Treating Persistent Corneal Epithelial Defects", submitted to Pharmaceutics MDPI. Your kind words regarding our work are deeply appreciated. Please, find attached the response to your comments and suggestions.
Once again, we are sincerely grateful for your time and the generous words you shared with us.

Reviewer 3 Report
Comments and Suggestions for Authors
The study investigates two formulations of compounded insulin eye drops, prepared at 1 IU/mL from commercial subcutaneous insulin, diluted in either saline solution or artificial tears. The primary focus is on their physicochemical characteristics, in vitro tolerance, and short-term stability under various storage conditions. The formulations were analyzed for isotonicity, pH levels, surface tension, and viscosity, demonstrating stable properties over 28 days when refrigerated.
Microbiological stability was confirmed, and insulin potency was maintained within the 90–110% range for the saline-based formulation over the specified period, whereas the artificial tears-based formulation showed a decrease in potency. Furthermore, in vitro testing suggested both formulations were well-tolerated by human and conjunctival cells, with the saline solution variant showing superior cell tolerance. Preliminary human data indicated the effectiveness and safety of the saline solution-based insulin eye drops for treating persistent corneal epithelial defects (PED), suggesting potential as an emergent therapy for such conditions.
The study offers potentially useful technical data that supports the utilization of insulin in the management of PED. The comprehensive examination of the physicochemical properties and stability of the insulin formulations adds valuable insights into their potential therapeutic benefits.
However, the conclusions drawn regarding the clinical efficacy and safety of the insulin eye drops, particularly the effectiveness in treating PED, are not fully substantiated by the presented results. I have some concerns for the authors to address.
Major Points:
1. The rationale for selecting a concentration of 1 IU/mL of Inclor for this study is not provided. Clarification on the choice of this specific dosage for the clinical trial is needed for a better understanding of its basis and implications.
2. The justification for employing Inclor in the clinical study is lacking. The authors must elucidate the reasons behind choosing Inclor for the intervention to establish its relevance and potential benefits in this context.
3. The manuscript does not clearly discuss the safety outcomes associated with Inclor use. It is crucial to detail these safety outcomes, their implications, and how Inclor fares in terms of safety to assess the risk-benefit ratio effectively.
4. The absence of information on a washout period raises concerns regarding the study's methodological rigor and the potential for carry-over effects, which could confound the results.
5. The methods section is vague regarding the statistical analyses applied, including whether analyses were conducted on both eyes of patients. A clearer explanation of the statistical approaches used is essential for the validation of the study's findings.
6. Given the study's design, utilizing subjects as their own control and the availability of alternative treatments, the claims of "excellent preliminary clinical results" for Inclor's efficacy and safety are not substantiated by the data. This overstatement needs to be addressed and aligned with the evidence provided.
Minor Points:
1. The manuscript would benefit from more precise language to enhance readability and comprehension.
2. The assertion that "topical insulin eye drops are a promising alternative PED treatment" requires backing by relevant literature. Incorporating citations would lend credibility to this
claim.
3. The acronym NSS should be defined upon its first use to ensure clarity for all readers.
4. The term ‘empty cells’ in tables 3 and 4 is ambiguous. If these represent a lack of statistical significance, as suggested, this should be explicitly stated to prevent misinterpretation of the data.
NA
Author Response
Dear Reviewer,
On behalf of all the authors, I would like to extend our sincere gratitude for the time and effort you dedicated to reviewing our research manuscript, "Topical Insulin Eye Drops: Stability and Safety of Two Compounded Formulations for Treating Persistent Corneal Epithelial Defects", submitted to Pharmaceutics MDPI. We recognize the considerable time and care you invested in the review process, and we are truly grateful for your thorough examination of our work.
Please, find attached a point-by-point response to your major and minor points. Once again, we would like to thank your revision work.

Round 2
Reviewer 3 Report
Comments and Suggestions for Authors
Thanks for the revision.